# Characterization of Enrichment Cultures of Anammox, Nitrifying and Denitrifying Bacteria Obtained from a Cold, Heavily Nitrogen-Polluted Aquifer

**DOI:** 10.3390/biology12020221

**Published:** 2023-01-30

**Authors:** Ekaterina Botchkova, Anastasia Vishnyakova, Nadezhda Popova, Marina Sukhacheva, Tatyana Kolganova, Yuriy Litti, Alexey Safonov

**Affiliations:** 1Winogradsky Institute of Microbiology, “Fundamentals of Biotechnology” Federal Research Center, Russian Academy of Sciences, 117312 Moscow, Russia; 2Frumkin Institute of Physical Chemistry and Electrochemistry, Russian Academy of Sciences, 117312 Moscow, Russia; 3Institute of Bioengineering, Research Center of Biotechnology, Russian Academy of Sciences, 117312 Moscow, Russia

**Keywords:** heavy nitrogen pollution, aquifer, low temperature, anammox bacteria, nitrifying bacteria, denitrifying bacteria, bioremediation, high nitrate buildup

## Abstract

**Simple Summary:**

Heavy nitrogen pollution is a major ecological problem for cold aquifers near the zone of a radioactive waste surface repository. Unique microbial communities are formed in these groundwater ecosystems that could be used for sustainable bioremediation in situ. Prior to field experiments, in this work we obtained and analyzed enrichment cultures of N cycle bacteria to shed light on their metabolic capabilities. Despite high N conversion activity, surprisingly, no “conventional” nitrifying bacteria were found in enrichment cultures. Moreover, stable nitrite production could be achieved through heterotrophic denitrification, which implies great potential to produce nitrite for sustainable anammox-based bioremediation. Anammox bacteria were difficult to enrich in fed-batch culture. However, stable removal of ammonium and nitrite under anaerobic conditions could be achieved in a continuous flow reactor packed with nonwoven fabric. Intriguing was the high nonstoichiometric buildup of nitrate, which could be an adaptive mechanism of anammox bacteria to low temperature. *Xanthomonadaceae* was a common taxon for all enrichment cultures, indicating its exclusive role in this ecosystem. Moreover, its representatives, genera *Thermomonas* and *Stenotrophomonas*, have nitrite oxidoreductases (nxr), which may explain the observed high accumulation of nitrates. These findings may be important in anammox-based low-temperature wastewater treatment.

**Abstract:**

Anammox bacteria related to *Candidatus* Scalindua were recently discovered in a cold (7.5 °C) aquifer near sludge repositories containing solid wastes of uranium and processed polymetallic concentrate. Groundwater has a very high level of nitrate and ammonia pollution (up to 10 and 0.5 g/L, respectively) and a very low content of organic carbon (2.5 mg/L). To assess the potential for bioremediation of polluted groundwater in situ, enrichment cultures of anammox, nitrifying, and denitrifying bacteria were obtained and analyzed. Fed-batch enrichment of anammox bacteria was not successful. Stable removal of ammonium and nitrite (up to 100%) was achieved in a continuous-flow reactor packed with a nonwoven fabric at 15 °C, and enrichment in anammox bacteria was confirmed by FISH and qPCR assays. The relatively low total N removal efficiency (up to 55%) was due to nonstoichiometric nitrate buildup. This phenomenon can be explained by a shift in the metabolism of anammox bacteria towards the production of more nitrates and less N_2_ at low temperatures compared to the canonical stoichiometry. In addition, the too high an estimate of specific anammox activity suggests that N cycle microbial groups other than anammox bacteria may have contributed significantly to N removal. Stable nitrite production was observed in the denitrifying enrichment culture, while no “conventional” nitrifiers were found in the corresponding enrichment cultures. *Xanthomonadaceae* was a common taxon for all microbial communities, indicating its exclusive role in this ecosystem. This study opens up new knowledge about the metabolic capabilities of N cycle bacteria and potential approaches for sustainable bioremediation of heavily N-polluted cold ecosystems.

## 1. Introduction

Low-temperature groundwaters provide a unique habitat for psychrotolerant and psychrophilic microbial communities. However, aquifers located near sludge repositories can contain high concentrations of nitrogen, which is a serious ecological problem nowadays [1]. Such polluted groundwaters favor the development of phylogenetically diverse and metabolically versatile communities of N-cycle microorganisms that are responsible for N_2_ formation.

In groundwaters, denitrifying bacteria are one of the most important microorganisms involved in nitrogen transformation [2,3,4,5]. Depending on the level of groundwater pollution and, accordingly, the availability of electron donors, denitrifiers can grow using various electron donors. For example, in oligotrophic aquifers where organic matter is absent, denitrifiers use sulfur compounds or hydrogen [6]. Aerobic ammonia oxidizers, including both bacteria and, to a lesser extent, archaea, also occur in aquifer communities, including oligotrophic and nutrient-rich groundwaters [7,8,9,10,11]. Other processes of nitrogen transformation in aquifer systems include n-DAMO (nitrite/nitrate-dependent methane oxidation), which, however, more rarely coexists with denitrification and anaerobic ammonium oxidation (anammox) [4], and respiratory ammonification [12]. Anammox bacteria that are capable of anaerobic ammonium oxidation with nitrite [13] play a significant role in the global N-cycle, producing up to 50% of gaseous nitrogen in natural habitats [14,15]. Nowadays, they are discovered in various ecosystems, including groundwaters [2,15,16]. In such habitats, especially those subjected to anthropogenic pollution (for example, as a result of intensive farming using mineral nitrogen-containing fertilizers), anammox bacteria may be the dominant microorganisms of the nitrogen cycle [17]. The contribution of anammox bacteria to the formation of molecular nitrogen can exceed 30% in such ecosystems [18,19]. Anammox bacteria in groundwaters are diverse, and most phylotypes belong to novel lineages in the phylum Planctomycetes [18,20]. In aquifers, they compete with denitrifiers for nitrite [21]. In the case of low organic carbon availability, anammox activity can be higher than that for denitrification, resulting in significantly higher impact of anammox bacteria on N_2_ production [2,22]. In heavily contaminated groundwaters, processes of nitrogen transformation co-occur with transformation of other pollutants. For instance, in Fe-rich waters, anaerobic ammonium oxidation can be coupled with iron reduction (“Feammox”) [23], and in cases of thiosulfate availability, denitrification can be linked to the oxidation of reduced sulfur compounds [6].

Anammox bacteria are known to be slow-growing microorganisms, especially at low temperatures, because they conserve energy for core metabolism due to a decrease in biosynthesis in general, which leads to a decrease in activity and growth rate [24]. Thus, enrichment and accumulation of active anammox biomass is challenging itself, especially in the case of low temperatures [25,26]. However, there is information about the activity of anammox bacteria in seawaters at 15 °С [27]. The optimum temperature for anammox activity in natural habitats may be as low as 9 °C in permeable Arctic sediment [28], 12 °C in the Arctic, permanently cold marine sediments [29], or 15 °C in marine sediment from Skaggerak (North Sea) [30]. For low-temperature wastewater treatment, mesophilic anammox bacteria can gradually be adapted to 12 °C and even 10 °C [31,32]. Application of anammox bacteria in wastewater treatment in regions with cold climates is complicated due to the limited knowledge and strategies of adaptation of anammox communities to low temperature, which can take months and even years. Anammox bacteria are also considered prospective for in situ bioremediation of complex polluted groundwaters found near storages of chemical wastes from mining and processing facilities [7,8,10].

In previous work, it was found that microbial communities from groundwaters polluted with uranium and nitrate from sludge storage at Chepetsky Mechanical Plant (CHMZ) (Glazov, Udmurtia, Russia) developed in situ at low temperatures typical for groundwaters. The microbial community included microorganisms involved in metal cation reduction and uranium immobilization, and bacteria, important for N-cycling processes: denitrifiers, nitrifiers and anammox bacteria. Anammox bacteria belong to the genus *Candidatus* Scalindua [33]. Anammox bacteria have a tendency for attached growth, and thus the anammox community requires specific carriers for biofilm formation [32,34]. Another important feature is that removal of oxidized and reduced forms of nitrogen in complex pollution of groundwaters near the sludge storage reduces the migration of metals, including actinides, in dissolved mobile forms, thus establishing a complex biogeochemical barrier in situ under polluted conditions [35,36,37].

The current study is a part of the complex research of groundwaters near CHMZ, a unique site where conditions established more than 50 years ago benefit the development of anammox bacteria. Groundwater pollution occurs due to filtration of dissolved components of the sludge with the flow of the groundwaters through protective walls of the storage, which degraded because the storage was put into operation more than 50 years ago. Thus, unique conditions for the functioning of various groups of N-cycling bacteria under low concentrations of organic carbon have been established. The major aim of the current study was to enrich the microbial community from the groundwater site in the laboratory conditions in order to reveal the anammox bacteria and other N-cycling microorganisms and assess their activity in the community. The characterization of the obtained enrichments and the reconstruction of metabolic pathways could provide valuable data for the potential use of this unique microbial community in various sustainable bioremediation approaches.

## 2. Materials and Methods

### 2.1. Sampling

Groundwater samples were collected at a depth of 12 m from an aquifer near the CHMZ sludge storage with ammonium, nitrate and sulfate contamination during pumping of 2 full volumes of the wellbore. The temperature of groundwater during the year does not undergo significant changes and fluctuates in the range of 8–10 °С. At the time point of sampling, the amount of ammonia ions reached 58.4 mg/L, nitrite 28 mg/L, nitrates 7400 mg/L, sulfate 1803 mg/L and bicarbonate ions 280 mg/L. The chemical composition of the groundwater is listed in Table 1. A low content of phosphates and organic carbon indicated oligotrophic conditions. The presence of nitrates, sulfates and other oxidizing agents likely explains the high redox potential of groundwater. This indicates that the activity of the anammox process is possible only in biofilms with an Eh gradient or after removal of nitrate due to denitrification, which leads to a decrease in the redox potential [34]. The groundwater sample was inhabited by the *Candidatus* Scalindua-dominated anammox community [33].

To analyze the microbial community, a groundwater sample was preserved with ethyl alcohol up to 30 wt. %. To isolate DNA, groundwater was passed through a cellulose acetate filter with a pore diameter of 0.22 μm (Vladipor, Vladimir, Russia). Groundwater for enrichment studies was sampled in sterile 1.5 L bottles to the top and stored at 4 °C until use.

### 2.2. High-Throughput Sequencing of 16S rRNA Genes

DNA was isolated using the FastDNA™ SPIN Kit for Soil (MP Biomedicals, Santa Ana, CA, USA) according to the manufacturer’s instructions. The preparation of amplicon libraries of the V4 region of the 16S rRNA gene was carried out as described previously [38] using a pair of primers: 515F (50-GTGBCAGCMGCCGCGGTAA-30 [39] and Pro-mod-805R (50-GGACTACHVGGGTWTCTAAT-30 [40]. The libraries were sequenced on a MiSeq system (Illumina, San Diego, CA, USA) using a 150-nucleotide length paired-end read cartridge. Libraries were prepared and sequenced into two replicas for each sample. The amplicon sequence variant (ASV) table was constructed using the Dada2 script [41] and the SILVA 138.1 database [42]. Analysis of the ASV table was performed using MicrobiomeAnalyst [43].

### 2.3. Fed-Batch Cultivation without Carrier Materials

Nitrifying bacteria were enriched on Winogradskiy medium at 15 °С [44], denitrifying bacteria were enriched on Adkins medium at temperature 15 °С [1]. Medium for stage I nitrifiers (aerobic ammonia oxidizers, AOB) contained (g/L tap water): (NH_4_)_2_SO_4_—2.0; K_2_HPO_4_—1.0; MgSO_4_ ∙ 7H_2_O—0.5; NaCl—2.0; FeSO_4_ ∙ 7H_2_O—0.05; CaCO_3_—5.0, рН = 8. Medium for stage II nitrifiers (nitrite oxidizing bacteria, NOB) contained (g/L tap water): NaNO_2_—1.0; K_2_HPO4—0.5; MgSO_4_ ∙ 7H_2_O—0.5; NaCl—0.5; FeSO_4_ ∙ 7H_2_O—0.4; Na_2_CO_3_—1.0, рН = 8. Medium for denitrifying bacteria (DB) contained (g/L): NH_4_Cl—1.0; KH_2_PO_4_—0. 75; K_2_HPO_4_—1.5; NaNO_3_—1.0; NaCl—0.8; Na_2_SO_4_—0.1; MgSO_4_·7H_2_O—0.1; KCl—0.1, yeast extract—0.5; glucose—1.0; CH_3_COONa—1.0, pH = 7, gas phase—Ar. The procedure for obtaining the AOB, NOB and DB enrichment cultures was carried out according to previous work [9]. AOB and NOB enrichment cultures were aerated at a very low rate (about 1 mL/min, through a medical needle connected to an aquarium pump).

The batch culture medium for anammox bacteria (AnAOB) was as follows [45] (in mg/L): NaNO_2_—222 (45 mg N-NO_2_/L), NH_4_Cl—172 (45 mg N-NH_4_/L), NaHCO_3_—800, MgSO_4_ ⋅ 7H_2_O—120, CaCl_2_ ⋅ 7H_2_O—180, KH_2_PO_4_—27, trace elements solution—0.5 mL/L (EDTA—5, H_3_BO_3_—0.014, СoCl_2_⋅ 4H_2_O—2, ZnCl_2_—0.203, MnCl_2_ ⋅ 4H_2_O—0.99, CuSO_4_ ⋅ 5H_2_O—0.25, (NH_4_)_6_Mo_7_O_24_ ⋅ 4H_2_O—1.24, NiCl_2_ ⋅ 6H_2_O—0.19, Na_2_SeO_3_—0.105). The fed-batch enrichment of anammox-bacteria was carried out in 230 mL glass bottles at 15 °C without stirring. Feeding was accomplished by decanting two-thirds of the medium and adding a new medium. The headspace of the vials was purged with argon and sealed with a rubber stopper and an aluminum crimp cap at the onset and after each feeding.

N consumption and production rates were calculated using Equation (1):(1)$$N_{\mathrm{rate}}=\frac{\left(N_{initial}-N_{final}\right)\times V_{medium}}{t}$$
where $N_{\mathrm{rate}}$—N-NH_4_ or N-NO_2_ consumption and N-NO_2_ or N-NO_3_ production rates, mg N/L/day; $N_{initial}$ and $N_{final}$—initial and final concentrations of N-NH_4_, N-NO_2_ or N-NO_3,_ mg N/L; $V_{medium}$—volume of the liquid phase, L; t—incubation time, days.

### 2.4. Batch Cultivation with the Addition of Carrier Materials

Because anammox bacteria (1) were found to be the least active microbial N-cycle group in fed-batch culture and (2) are known to prefer attached growth, batch culture was also performed with the addition of several popular carrier materials for biomass immobilization and biofilm formation. Briefly, 1 g of the carriers (see Table 2) was placed into each sterile 30 mL glass vial together with 20 mL of groundwater. In groundwater, the concentration of ammonium and nitrite nitrogen was preliminarily set at 50 mg N-NH_4_/L and 60 mg N-NO_2_/L by adding the appropriate amount of NH_4_Cl and NaNO_2_. The number of anammox cells (according to direct counting of cells hybridized with a Cy-3-labeled anammox-specific FISH probe) in groundwater was approximately 2 × 10^4^ cells/mL. The glass vials were purged with argon and sealed with a rubber stopper and an aluminum crimp cap. The experiment was carried out in duplicate. Incubation was carried out without stirring in the dark at 15 °C for 35 days. Samples for the determination of ammonia, nitrite, and nitrate nitrogen, pH, and Eh were taken every 5 days.

### 2.5. Continuous Flow Cultivation of AnAOB

After finding the most suitable carrier for the anammox community according to fed-batch experiments, continuous flow cultivation was performed for assessment of the long-term nitrogen removal activity. The setup of the anammox biofilm reactor (ABR) is shown in Figure 1. The ABR was a 0.5 L glass bottle with a 0.33 m × 0.66 m nonwoven fabric for biomass immobilization. The synthetic medium was supplied to the ABR reactor using a peristaltic pump (once per hour in equal portions of the medium) according to Table 3. The synthetic medium was modified from previous works [46,47] (mg/L): NH_4_Cl and NaNO_2_ according to Table 3, NaHCO_3_—1000, MgSO_4_ ⋅ 7H_2_O—120, CaCl_2_ ⋅ 7H_2_O—180, KH_2_PO_4_—27, trace elements solution—0.5 mL/L (EDTA—5, H_3_BO_3_—0.014, СoCl_2_⋅ 4H_2_O—2, ZnCl_2_—0.203, MnCl_2_ ⋅ 4H_2_O—0.99, CuSO_4_ ⋅ 5H_2_O—0.25, (NH_4_)_6_Mo_7_O_24_ ⋅ 4H_2_O—1.24, NiCl_2_ ⋅ 6H_2_O—0.19, Na_2_SeO_3_—0.105), folic acid—25 [48], formic acid—0.025 [48]. pH of the synthetic medium was 7.5–8, which is optimal for the growth of anammox bacteria [49]. Every 3–4 days, a fresh synthetic medium was prepared, which was thoroughly purged with argon and stored in a 5 L influent glass container under argon (using a gas bag). ABR was stored in a thermostated cabinet at a temperature of 15 °C. The nitrogen load was gradually increased up to the 180th day (phases 1–3), and then, after the deterioration of nitrogen removal, it was reduced (phase 4), according to Table 3.

### 2.6. Analytical Methods

Characteristics of the porous structure of the material samples were studied by the method of low-temperature nitrogen adsorption measurement. The specific surface area of the materials was estimated by the BET method [50]. The concentrations of anions and cations were measured on a Capel-205 new generation capillary electrophoresis system (Lumex, St. Petersburg, Russia). Electrophoresis was performed in untreated fused-silica capillaries of 60 cm length and 75 μm internal diameter. The capillary was held at 20 °C and the applied voltage was 13 kV for cations or −17 kV for anions [34]. The pH and Eh were determined using an FE20 pH meter equipped with an InLab^®^ microelectrode and ORP electrode InLab Redox Micro (all Mettler Toledo, Greifensee, Switzerland). Development rates of microbial communities on carriers were determined using the MTT assay [34].

### 2.7. Fluorescence In Situ Hybridization (FISH)

FISH analysis was performed to estimate the presence and relative abundance of various microbial groups and their dynamics. Hybridization was carried out at a temperature of 46 °С with 16S rRNA-specific Cy-3-labeled oligonucleotide probes (“Syntol”, Moscow, Russia) according to the standard scheme [51], with modifications [52,53]. All of the probes, together with their specificity, are listed in Table 4. Samples were analyzed using an epifluorescence microscope Leitz (Wetzlar, Germany) with Zeiss 20 filter for Cy3-labeled probes, with a digital camera, Nikon DS-Fi1c (Tokyo, Japan) at 100 × 10 magnification.

### 2.8. Visualization and Analysis of the Microbial Biofilms

Visualization and analysis of the microbial biofilms were performed using a Leica SP5 confocal scanning laser microscope (Leica, Wetzlar, Germany) and program package Comstat 2.1 for ImageJ. All polymeric carriers were washed with distilled water to eliminate planktonic cells before biofilm staining. Staining of the polysaccharide matrix was performed with lectin concanavalin A (conA), conjugated with Alexa Fluor-488 (C11252, Thermo Fisher, Waltham, MA, USA) for 30 min in dark. After staining, the samples were washed with distilled water and visualized at 20× magnification. The surface of the biofilm was calculated using ImageJ 1.52 with program package COMSTAT 2.0. The microscopic analysis was done using equipment of the Core Centrum of the Institute of Developmental Biology RAS.

### 2.9. Quantitative Polymerase Chain Reaction (qPCR)

In continuous-flow enrichment culture, a mixed sample (attached and suspended) from the last phase was used for DNA isolation. DNA was isolated from the samples using a MoBio Power Soil DNA Isolation Кit (MoBio Laboratories, Berlin, Germany) according to the manufacturer’s instructions. Real-time PCR was carried out at PCR buffer-RT (Syntol, Moscow, Russia) in the presence of SYBR Green I stain and passive reference stain ROX for normalization of the fluorescence of the stain used in the reaction. Detection for each sample was carried out in duplicate. ddH_2_O (Syntol, Moscow, Russia) was used as a negative control (reaction mixture without a DNA template). Amplification was performed using the CFX96 Touch™ Real Time PCR Detection System (Bio-Rad, Hercules, CA, USA). To calculate the number of copies of the gene in the analyzed sample, the obtained signal was compared with a standard curve, which was constructed using a series of serial dilutions of the standard sample. The target PCR fragment, preliminarily purified using the WizardSV Gel and PCR Clean-Up System (Promega, Madison, WI, USA) and subsequently cloned into the pGEM-T vector (Promega, Madison, WI, USA) was used as a standard sample.

To determine the presence of anammox bacteria, primers targeting the 16S rRNA gene were used: А438F/А684R (5′-GTCRGGAGTTADGAAATG-3′/5′-ACCAGAAGTTCCACTCTC-3′)’ [59]. Temperature profile of the reaction was the following: polymerase activation 5 min at 95 °С, next 40 cycles–30 s at 95 °С, 15 s at 56 °С and 45 s at 62 °С. The amount of AOB were determined using primers CTO 189fA/B (5′-GGAGRAAAGCAGGGGATCG-3′), CTO 189fC (5′-GGAGGAAAGTAGGGGATCG-3′) in ratio 2:1 and RT1r ‘(5′-CGTCCTCTCAGACCARCTACTG-3′) according to the previously published protocol [60]. The number of the copies of nitrate reductase genes (*nirK* and *nirS)* were determined using the following primers: nirS1R/nirS3R (5′-CCT AYT GGC CGC CRC ART-3′/5′-GCCGCCGTC(A/G)TG(A/C/G)AGGAA-3′), nirK1F/nirK5R (5′-GGM ATG GTK CCS TGG CA-3′/5′-GCC TCG ATC AGR TTR TGG-3′) according to the previously published protocol [61]. DNA was isolated from the sediment obtained after the centrifugation of the initial sample. The resulting precipitate was 55 ng. The amount of isolated DNA was 4385 ng.

## 3. Results

### 3.1. Enrichment Cultures of N-Cycle Bacteria

#### 3.1.1. N Consumption/Production Rates in Fed-Batch Cultures

Figure 2 shows rates of nitrogen consumption during three consecutive feedings of AOB, NOB, DB and AnAOB with corresponding mediums. All three feedings in the AOB enrichment study had more or less the same nitrogen production and consumption. During the last feeding, the rate of ammonium consumption by AOB was 38.4 mg N/L/day and the rate of nitrite formation reached 28.4 mg N/L/day. It should be noted that nitrate accumulation was also observed. For the last feeding, it was 10.2 mg N/L/day. Thus, during fed batch cultivation of AOB on Winogradsky’s medium, two stages of nitrification were presumably observed.

During fed-batch enrichment of NOB, levels of nitrite intake and nitrate production steadily increased in subsequent feeding stages (Figure 2). Thus, the maximum rates of nitrite removal (16.1 mg N/L/day) and nitrate formation (11.3 mg N/L/day) were observed during the third feeding.

The potential denitrifying activity of the microbial community of the groundwater ecosystem was also quite high and was estimated at 9.1–11.6 mg N-NO_3_/L/day. Interestingly, however, an increase in nitrite formation was observed at the following feedings (up to 9.47 mg N-NO_2_/L/d at the third feeding), suggesting the potential for nitrite formation for the AnAOB community.

AnAOB activity was found to be the lowest among all the N cycle processes studied in this work. The maximum consumption rates of ammonium (0.63 mg N-NH_4_/L/day) and nitrite (0.59 mg N-NO_2_/L/day) were recorded during the first feeding. The rate of nitrogen consumption at the second feeding was 4.2–4.8 times lower, and at the last feeding it was zero. This observation may suggest that the AnAOB enrichment conditions were not optimal.

#### 3.1.2. Results of FISH Analysis of AnAOB Enrichment Culture

Since the activity of anammox bacteria decreased during batch cultivation, a FISH analysis was performed to check the dynamics of the relative abundance of anammox bacteria in the AnAOB enrichment culture. Results of hybridization with Cy-3-labeled probes targeting anammox bacteria are shown in Figure 3. During the experiment, a decrease in the number of anammox cells was observed from 15–20% of the total number of microbial cells at the beginning of the first feeding to 5% at the end of the third feeding. Coccoid cells formed clusters of cells embedded in an extracellular polymeric matrix, which is typical for anammox bacteria.

#### 3.1.3. Microbial Diversity at the Family Level in Enrichment Cultures

The results of profiling the 16S rRNA gene of enrichment cultures at the end of the third feeding are shown in Figure 4. Analysis of the microbial diversity of the AOB enrichment culture showed the dominance of *Pseudomonadaceae* and *Rhodanobacteracea*. “Canonical” nitrifiers were not found in the microbial communities or in the original groundwater [33]. However, the NOB enrichment culture contained ammonium-oxidizing archaea (AOA) of *Crenarchaeota* (0.5% of phylotypes belonged to the family *Nitrososphaeraceae*—Ca. *Nitrocosmicus* [62]. Also, methanogenic archaea (*Euryarchaeota*—family *Methanobacteriaceae* [63]) and ferrum-oxidizing archaea (*Thermoplasmatota*—family *Ferroplasmaceae* [64]) were found. Heterotrophic nitrifying bacteria also capable of aerobic denitrification predominated in the AnAOB, AOB and DN enrichment cultures. They belonged to the families *Pseudomonadaceae* [65], *Rhodanobacteraceae* [66], *Xanthomonadaceae* [65], *Alcaligenaceae* [67]. These same groups are known for their ability to complete or partial denitrification under anaerobic conditions.

### 3.2. Continuous Flow Cultivation of AnAOB

#### 3.2.1. Anammox Performance with Various Carriers

In Figure 5 and Figure 6, data on nitrogen consumption, respiratory activity, and the total area of polysaccharides in biofilms on various carriers during batch cultivation of the microbial community of groundwater in an anammox medium are presented. The maximum biofouling was observed on polymer brushes and nonwoven fabric. A relatively low area of biofouling was on the rice husk biochar, polyester felt, and kaldness. At the same time, the maximum respiratory activity was recorded on polymer brush, fiberglass mesh and two carriers from HelX (kaldness and Flake 30). However, it should be noted that due to the slow growth of anammox bacteria, biofouling does not always correspond to a high content of anammox bacteria, but is often caused by rapidly developing heterotrophic bacteria [33].

Nitrite consumption was the fastest in batches with the addition of polyester felt and kaldness. For carrier materials other than fiberglass mesh, the average nitrite nitrogen uptake rates were close to each other. At the same time, the most efficient consumption of ammonium was observed in batches with the addition of carbon felt and nonwoven fabric. Based on the totality of the studied parameters and some positive experiences of other researchers [68,69,70], a nonwoven fabric was chosen as a carrier for continuous cultivation of AnAOB in the ABR.

#### 3.2.2. N Removal Performance in ABR

After the addition of nonwoven fabric as a carrier and inoculation by groundwater, ABR was subsequently operated for a period of 285 days (Figure 7, Table 3). Four main phases can be identified throughout this long experimental period according to N loading rate and N removal performance.

The first phase was the startup period. The N-NH_4_ concentration was gradually increased from 5 to 15 mg/L and the N-NO_2_ concentration from 5 to 25 mg/L. The influent flow rate was 122.4 mL/day, which corresponds to hydraulic retention time (HRT) of approximately 3.67 days. During days 1–30, ammonium nitrogen removal was 95–100%, nitrite nitrogen removal—50–90%, and total nitrogen removal 70–90% (Figure 6). On days 30 to 60, the system reached the stable efficiency of total N removal up to 98% (ammonium and nitrite nitrogen up to 100%). Therefore, the influent concentration of N-NH_4_ and N-NO_2_ was increased up to 45–50 mg/L and 35–40 mg/L, respectively. At the same time, efficiency of total nitrogen removal remained at the high level of 80–95% on days 60 to 80.

In the second phase of the experiment, the concentrations of ammonium and nitrite nitrogen in the influent remained unchanged, but the influent flow rate was doubled (244.8 mL/day). These conditions lasted 25 days. During this period, ammonium and nitrite nitrogen were also effectively removed, up to 99%. However, a decrease in the removal of total nitrogen from 80–95% to 60–70% was observed compared to phase I. This was explained by the increase in the concentration of N-NO_3_ in the effluent from 2 to 30 mg/L (Figure 7A).

In phase 3, the influent flow rate was doubled to 489.6 mL/day (HRT~0.92 days) to further increase the N removal rate. Under these conditions, the system was operated for about 70 days, the degree of N-NH_4_ and N-NO_2_ removal remained stable (95–100%). However, the production of N-NO_3_ continued to increase sharply from 30% to 70% of the loaded nitrogen, while the removal of total nitrogen decreased from 60% to 25% (Figure 7B).

Due to the low total nitrogen removal and the buildup of nitrate nitrogen in wastewater, in phase 4 of the experiment, the N loading rate was reduced by bringing the influent flow rate to 244.8 mL/day and reducing the concentration of ammonium nitrogen in the influent to 35–40 mg/l. Under such conditions, total nitrogen removal increased to 40% and formation of nitrate nitrogen was decreased by 5% in 28 days of operation. Since ammonium and nitrite nitrogen were consumed up to 100%, the concentration of ammonium nitrogen was increased to 75–85 mg/l, and nitrite nitrogen to 60–70 mg/l at a constant flow rate of the influent (Table 3). Under these conditions, the system operated for about 80 days, resulting in an increase in total nitrogen removal efficiency of up to 55%, but the concentration of nitrates in the effluent remained quite high (~70 mg N-NO_3_/L).

### 3.3. Key N Cycle Bacteria in the ABR Microbial Community

#### 3.3.1. Fluorescence In Situ Hybridization

Dynamics of the relative abundance of key N cycle bacteria in ABR is shown in Figure 8. The amount of anammox bacteria changes over time. A drop in the abundance of anammox bacteria during phase 3 of the continuous flow cultivation corresponds to reduction of anammox activity and possible inhibition of anammox bacteria, with subsequent restoration of activity and abundance of the physiologically active anammox cells. By the fourth phase of the experiment, the relative abundance of anammox bacteria reached 10% of all microbial cells in the reactor (Figure 8G). According to microscopy, anammox bacteria comprise densely packed clustered coccoid cells about 1 µm in diameter (Figure 8A,B). Such morphology is typical for this microbial group. Thus, though the relative amount of anammox cells was reduced during phase 3 (which is confirmed by the drop of the total nitrogen removal in the ABR), in the last phase the abundance and activity of anammox bacteria remained stable, which is confirmed by the results of qPCR.

Aerobic ammonium oxidizers (AOB) increased in number during the course of the experiment, reaching the highest amount by the fourth phase of the experiment (Figure 8C,D,G).

The presence of nitrite oxidizers (NOB) was also estimated using FISH analysis. Members of the genus Nitrospira were not found in the initial phases of the experiment; however, their relative abundance in phase 4 of the experiment reached 10% (Figure 8E,F,G).

#### 3.3.2. qPCR

qPCR analysis was performed only once (due to technical issues), at the end of phase 4 of continuous flow cultivation. Appendix A shows the results of qPCR analysis for various members of the N cycle microbial community in the ABR. According to the results, in phase 4 of the ABR operation, the most abundant were nitrifying bacteria, the least abundant anammox bacteria. Denitrifiers were represented by both nirK and nirS denitrifiers, and the number of nirK denitrifiers was significantly higher (by about three orders of magnitude).

#### 3.3.3. Microbial Community in ABR

According to the results of 16S rRNA gene sequencing, the microbial composition of the ABR is typical for the communities of wastewater treatment reactors of various sizes (laboratory and full-scale). The microbial community includes various phylotypes of microorganisms involved in nitrogen transformation (Figure 9). AOB were represented by autotrophic (*Nitrosomonas*) and heterotrophic (*Shinella*) genera [71]. Among potential DB were members of the families *Rhodocyclaceae*, *Comammonadaceae*, *Burkholderiaceae* and *Xanthobacteraceae* [72,73,74]. Nitrogen-fixing microorganisms include representatives of the family *Beijerinckiaceae*, order *Rhizobiale* [75]. Family *Ignavibacteriaceae*, which often coexists with AnAOB and DB, could also be involved in the nitrogen transformation or/and polymer degradation [76]. *Bacteroidetes*, an essential part of communities of nitrogen-contaminated sites were represented by the families *Chitinophagaceae* and *Microscillaceae* [73,77]. In the microbial community, 2.6% of all phylotypes belonged to the family *Pirellulaceae* of phylum *Planctomycetes*. However, phylotypes of “classical” anammox bacteria, i.e., families *Scalinduaceae* and *Brocadiaceae*, were not detected.

## 4. Discussion

Enrichment of psychrotolerant/psychrophilic anammox bacteria in laboratory conditions is a challenging and time-consuming procedure [25,26]. An attempt to enrich anammox bacteria by batch-fed culture at low temperature was unsuccessful. Anammox activity disappeared by the third feeding, probably due to the suboptimal conditions and/or the accumulation of potentially inhibiting metabolites in the medium. Many bacterial species have been found to exist in a viable but unculturable state [78,79] or otherwise in a “not immediately culturable” state, which can be attributed to the case of enrichment of anammox bacteria from a cold groundwater ecosystem in this work. However, certain success has been achieved during the long-term continuous flow cultivation in the ABR system packed with a nonwoven fabric. By phase 4 of the ABR operation, the efficiency of the removal of ammonium and nitrite under anaerobic conditions indicated a stable course of the anammox process in the ABR. The presence and activity of anammox bacteria and other microorganisms capable of nitrogen transformation were registered at all phases of ABR operation according to FISH with specific oligonucleotide probes. In ABR, dissolved ammonium and nitrite are converted to dinitrogen gas in the coupled processes of anammox and denitrification. In phase 4 of the ABR operation, these results were confirmed by the qPCR analysis, which also helped to estimate the abundance of anammox, denitrifying and nitrifying bacteria in the community. The abundance of anammox bacteria amounted to (2.41 ± 0.11) × 10^2^ gene copies/mL. This result was lower than previously reported for anammox bacteria from low-temperature bioreactors [31], which may mean that the conditions in the ABR do not sufficiently benefit the growth of anammox bacteria, so anammox bacteria are outcompeted by denitrifiers. However, the main reason for the observed differences may be that Hu et al. [31] adapted the “warm” anammox biomass to cold using a medium of the same composition. Due to the large distance between ABR and the groundwater well, synthetic medium was used in this work for continuous-flow culture instead of the original groundwater.

The level of AOB in the microbial community of ABR was relatively high, thus reflecting that even the trace amounts of oxygen in the synthetic medium (remaining after purging the medium with argon and its storage in an argon atmosphere) was enough for growth of aerobic nitrifiers. The presence of significant amounts of nitrifiers in bioreactors operating under microaerophilic and even anaerobic conditions have already been reported [2,80,81,82,83] The possible explanation to this fact is connected with the high-affinity terminal oxidases in AOB and NOB that makes it possible for them to thrive under low concentration of oxygen [84,85]. In some cases, nitrifiers can develop an additional anaerobic metabolism called nitrifier denitrification, which is related to formation of nitric oxide at anaerobic or microaerobic conditions [59,69]. However, the possible presence of this mechanism in the studied microbial community requires further investigations.

Both nirK and nirS denitrifiers were found in the microbial community. The nirS denitrifiers were more abundant ((1.07 ± 0.12) × 10^8^ copies per mL). The same distribution (prevalence of nirS denitrifiers over nirK) was shown previously in bioreactor communities [86,87]. Even though both genes, nirK and nirS, are responsible for the same reaction, they are found in different organisms, and variations in abundance might reflect different responses of these organisms to the environmental factors and substrate concentrations [86]. Another possible reason is lack of Cu, which is the key cofactor of nirK enzyme, in the medium [88].

Results of qPCR correlate with the results of FISH analysis. However, the amount of anammox bacteria estimated by FISH was one order of magnitude higher than that for qPCR. Presumably, anammox bacteria which naturally occur in densely packed clusters, could be overestimated during the direct count on the microscope when analyzing FISH probes.

Denitrifiers are not only competitors of nitrite to anammox bacteria. On the contrary, some of the denitrifying bacteria can supply anammox bacteria in biofilms with nitrite, which accumulates in the biofilm matrix during long-term cultivation [89]. Thus, the relationships between these two groups in the community might be more complex than expected. At the same time, the increasing abundance of NOB in the bioreactor community (according to FISH) is a matter of concern: it was reported previously that NOB can outcompete anammox bacteria in systems operating at low temperatures [90,91]. In this case, in systems with partial nitrification–anammox, establishing specific conditions preferable for anammox is needed, i.e., prevention of formation of small granules and reduction of aeration [26,92]. In phase 4 of the ABR operation, 16S rRNA gene profiling of the microbial community helped to estimate that it was diverse and included various members of N cycle processes and other groups that are typically found in similar communities. Though the abundance of phylum *Planctomycetes* was 2.4%, “canonical” anammox bacteria of the families *Brocadiaceae* and *Scalinduaceae* were not detected. This misrepresentation can be explained by the low abundance of anammox bacteria, which, however, could be traced using qPCR and FISH.

It is interesting to note that the almost complete conversion of ammonia and nitrite nitrogen in ABR operating under anaerobic conditions was accompanied by a high buildup of nitrate nitrogen (Figure 6A). According to the anammox stoichiometric equation (Equation (2)) [93] it follows that 0.26 M N-NO_3_ should be obtained as a result of the conversion of 2.32 M total N (1 M N-NH_4_ and 1.32 M N-NO_2_) fed.
1NH_4_+ + 1.32NO_2_^−^ + 0.066HCO_3_^−^ + 0.13H^+^ **→** 1.02N_2_ + 0.26NO_3_- + 0.066CH_2_O_0.5_N_0.15_ + 2.03H_2_O(2)

Taking late phase 4 of the ABR operation (Figure 7A) as an example, it can be seen that about 5.0 M N-NO_3_ was produced from the conversion of 10.35 M total N (5.35 M N-NH_4_ and 5.0 M N-NO_2_). Thus, the simplified hypothetical stoichiometry for nitrogen conversion at late phase 4 of the ABR operation will be as shown in Equation (3):1NH_4_ + 0.93NO_2_ + 0.93H_2_O **→** 0.5N_2_ + 0.93NO_3_ + 5.86H^+^(3)

Considering (1) the absence of active aeration in the ABR and (2) the adoption of the maximum 1 mg O_2_/L as the trace concentration of dissolved oxygen (DO) in the synthetic medium after all manipulations to ensure anaerobiosis, the maximum concentration of nitrate nitrogen in the effluent can be 0.015 mM N-NO_3_/L per 5.35 mM N-NH_4_ consumed from aerobic ammonium oxidation (according to Equations (4) and (5)) or 0.063 mM N-NO_3_/L per 5.0 mM N-NO_2_ consumed from aerobic nitrite oxidation (according to Equation (5)).
1NH_3_ + 1.5O_2_ **→** 1NO_2_^−^+ 1H^+^ + 1H_2_O(4)
1NO_2_^−^ + 0.5O_2_ **→** 1NO_3_^−^
(5)

The quantities of nitrate that can be produced from the remaining dissolved oxygen are much smaller than those observed in this system, which means that most of the nitrate was produced anaerobically from the conversion of ammonium and nitrite. The observed interesting phenomenon can be explained by a shift in the metabolism of anammox bacteria towards the production of more nitrate and less gaseous N_2_ at low temperatures compared to the canonical stoichiometry (Equation (2)) calculated from the experiment in a sequencing batch reactor at 32–33 °C [93]. However, since the original microbial community of groundwater is unique, some syntrophic interaction of different microbial groups may have been established to also carry out the anammox process. A similar observation of high nitrate production in the anammox reactor was reported by Li et al. [85], who attributed this phenomenon to low N loading rates (0.8 kg N/m^3^/day), resulting in low N removal efficiency (<60%) due to nitrate accumulation. Metagenomic analysis indicated relatively higher levels of reads from genes predicted to be nitrite oxidoreductases (nxr) in the high nitrate producing anammox reactor. Also, both a decrease in anammox bacteria and an unexpected increase in nxr suggested that the higher nitrate production was not solely due to obligate NOB, but other nxr-containing bacteria are important contributors as well [85].

It is very interesting to compare the N removal rates per gram of volatile suspended solids (VSS) of anammox biomass reported in the literature and estimated in this work. The N removal rate can range from 0.0024 to 0.638 g N/g VSS/day, depending on the origin of the wastewater, temperature, N loading rate, etc. [94,95,96,97,98,99]. Considering an average removal of 55 mg N/L/day in late phase 4 and an estimate of 0.1–0.3 g biomass-VSS/L in ABR, the removal rate of 1.8–5.5 g N/g VSS/day seems too high for a purely anammox process. This means that not only anammox but also other microbial groups of the nitrogen cycle contributed greatly to nitrogen removal. The elucidation and in-depth study of these nitrogen-removing microbial groups should be one of the directions of future work.

Overall, fed-batch enrichment of AOB, NOB, and DB was more successful than AnAOB, with N conversion rates being an order of magnitude higher on average than AnAOB (Figure 2). With the enrichment of DB with each subsequent feeding, an increase in the ability for partial denitrification with the formation of nitrites was shown. This could be explained by the activity of organotrophic microbiota (*Carnobacteriaceae*) and members of *Alcaligenaceae* capable of assimilatory nitrate reduction to nitrite [100,101]. The potential for stable nitrite production through partial denitrification route is a viable option for sustainable anammox-based bioremediation in this cold nitrogen-polluted aquifer. It should be noted, that the composition of organotrophic denitrifying enrichment cultures, grown on acetate and glucose cannot be directly compared with the composition of the microbial community of groundwater. However, for the removal of nitrates in situ and the creation of anaerobiosis to start anaerobic processes by injecting soluble cheap simple electron donors (acetate, sugar) into the wells, as was done earlier by us [1] and other researchers [102,103], the data obtained may be useful for a preliminary assessment of the most active organotrophic denitrifies. It should also be added that representatives of the *Pseudomonadaceae* [65], *Rhodanobacteraceae* [66], *Xanthomonadaceae* [65], and *Alcaligenaceae* [67] families found in enrichment culture are standard representatives of the reduction branch of the nitrogen cycle in groundwater. In the groundwater sample, the microbial composition of which is given in the previous work [33], bacteria of the *Pseudomonadaceae* family found in a denitrifying enrichment culture predominated.

Obtained enrichment of AOB and NOB carried out nitrification processes, although no “classical” nitrifiers were found as a result of 16S rRNA profiling of cultures. The common taxon for the microbial communities of the enrichment cultures and the ABR, according to the Venn diagram (Figure 10), is the *Xanthomonadaceae* family.

This is a fairly large group capable participating in both the oxidation and reduction branches of the nitrogen cycle [65,104]. In the *Xanthomonadaceae* family, according to an analysis of 12 known strains in the KEGG database, an assimilation pathway for the reduction of nitrate to ammonium was found. In this case, the accumulation of nitrite may not be as significant due to the lower rate of the process. In the presence of high concentrations of nitrate, this feature alleviates toxic stress of nitrite, in contrast to classical denitrifiers. In addition, Fitzgerald et al. [105] reported the involvement of *Luteibacter* spp. of *Xanthomonadaceae* family in autotrophic utilization of ammonia as a sole source of nitrogen in the low DO-nitrification process. Based on preferential enrichment, and the reports from existing literature, organisms belonging to the family *Xanthomonadaceae* have the potential to participate in ammonia as well as nitrite oxidation, can function either as heterotrophic nitrifiers, or via autotrophic nitrification through yet uncharacterized pathways [106]. *Pseudoxanthomonas* of the family *Xanthomonadaceae* was also reported to be among the dominant denitrifiers in enrichments from brackishwater ecosystems [107]. Moreover, representatives of *Xanthomonadaceae*, genera *Thermomonas* and *Stenotrophomonas*, have nitrite oxidoreductases (nxr), which may explain the observed high accumulation of nitrates [85]. The *Nitrosomonadaceae* family occurs only in the AnAOB enrichment culture and ABR communities. The NOB and ABR communities have the largest number of unique representatives. Most representatives of the AOB enrichment culture are also found in the NOB community, which demonstrates the highest level of biodiversity and unique representatives.

Appendix A–E presents hypothetical metabolic pathways built using the program IVICODAK [107]. Major metabolic pathways in dominant microbial groups are disclosed in Appendix A. Pathways of anaerobic ammonium oxidation (anammox), denitrification and nitrification were found in ABR, which is confirmed by qPCR and FISH. Also, pathways of nitrogen fixation and nitrate reduction were found. Interestingly, both enrichment cultures of denitrifiers (DB) and fed-batch cultures of anammox bacteria (AnAOB) have denitrification pathways that correlate with the results of profiling the 16S rRNA gene in these enrichment cultures. In DB enrichment culture, the anammox pathway was absent, presumably, due to the presence of organics which could be toxic to anammox bacteria. According to the analysis, the ABR community contained the highest number of metabolic pathways, which belonged to different processes of the nitrogen cycle, including enzymes for all steps of the anammox process, unlike the AnAOB enrichment culture. These results support our observation that batch culture is not suitable for the enrichment of anammox bacteria. Continuous flow culture allows establishing conditions for growth of metabolically and phylogenetically versatile microbial communities, thus enabling their stable functioning over time and under changing environmental conditions.

## 5. Conclusions

In the present study, enrichment cultures of various N cycle microorganisms were obtained from a unique low-temperature aquifer heavily polluted with nitrate and ammonium. The absence of “conventional“ nitrifier phylotypes indicates that nitrification in this polluted ecosystem may have proceeded in a nonstandard way and that “conventional” nitrification may have disappeared over time due to high N load. The stable nitrite production by denitrifying bacteria suggests the feasibility of sustainable anammox-based bioremediation. Fed-batch culture was not suitable for enrichment of anammox bacteria, probably due to its unculturable or “not immediately culturable” state in response to unfavorable circumstances. However, a microbial community containing anammox bacteria in continuous flow culture could achieve stable ammonium and nitrite removal (up to 100%) under anaerobic conditions. Though the number of anammox cells was relatively low, up to 55% of total N was removed. Observed nonstoichiometric nitrate buildup can be explained by a shift in the metabolism of anammox bacteria towards the production of more nitrate and less gaseous N_2_ at low temperatures than the canonical stoichiometry of the anammox process. Moreover, the too high an estimate of the specific anammox activity suggests that N cycle microorganisms other than anammox bacteria may have contributed significantly to N removal. The elucidation and in-depth study of these N-removing microbial groups should be one of the directions of future work.

Taken together, this study contributes to gaining new knowledge about the metabolic capabilities of cold groundwater ecosystems. The potential ability of partial denitrification, robustness of the community in unstable and unfavorable environmental conditions, tolerance to high amounts of pollutants, high level of microbial diversity and activity of key N cycle microbial groups makes the studied community perspective for in situ bioremediation of contaminated groundwater. At the same time, the direct connection of processes in enrichment cultures with the natural system should be treated with caution. Future efforts should be aimed at shedding light on the metabolism of anammox bacteria at low temperatures. Particularly, high nonstoichiometric nitrate buildup is an interesting phenomenon, which can be a serious operational concern for the application of anammox technology to low-temperature wastewater treatment.

## Figures and Tables

**Figure 1 biology-12-00221-f001:**
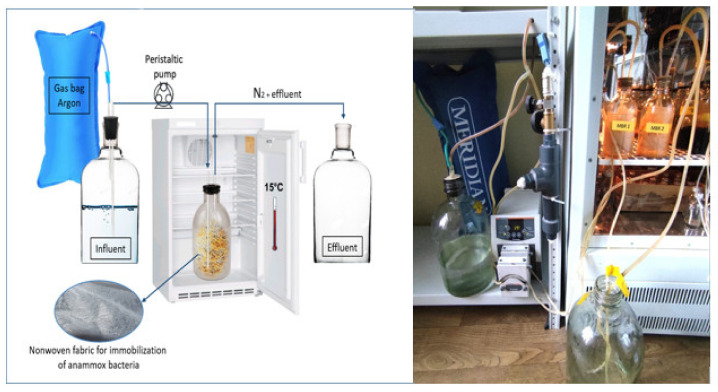
Schematic (**left**) and general (**right**) view of the continuous-flow ABR.

**Figure 2 biology-12-00221-f002:**
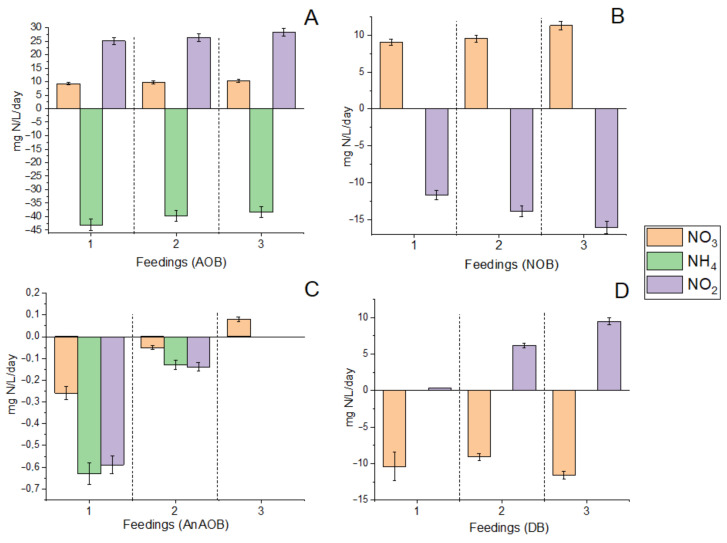
N consumption (−) and production (+) rates in enrichment cultures of AOB (**A**), NOB (**B**), AnAOB (**C**) and DB (**D**).

**Figure 3 biology-12-00221-f003:**
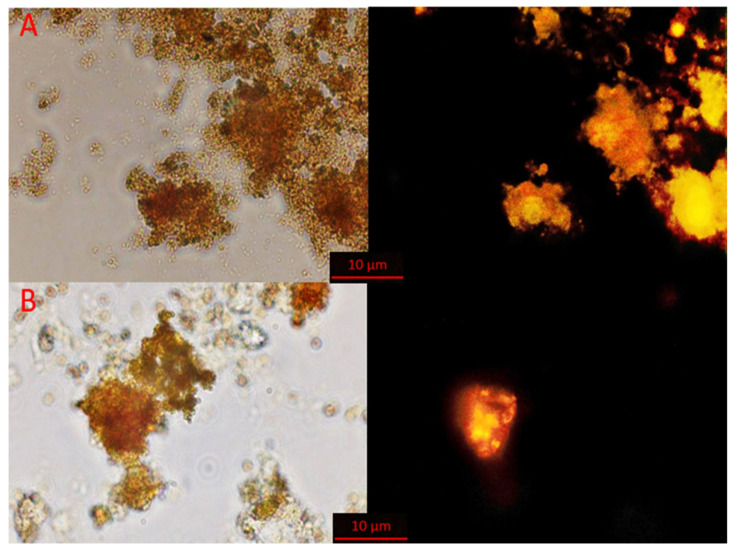
Fluorescence in situ hybridization (FISH) micrograph of biomass samples from AnAOB enrichments at the beginning of the first feeding (**A**) and at the end of the third feeding (**B**). Left—phase contrast, right—hybridization with oligonucleotide probe amx368 targeting anammox bacteria. Scale bar 10 µm.

**Figure 4 biology-12-00221-f004:**
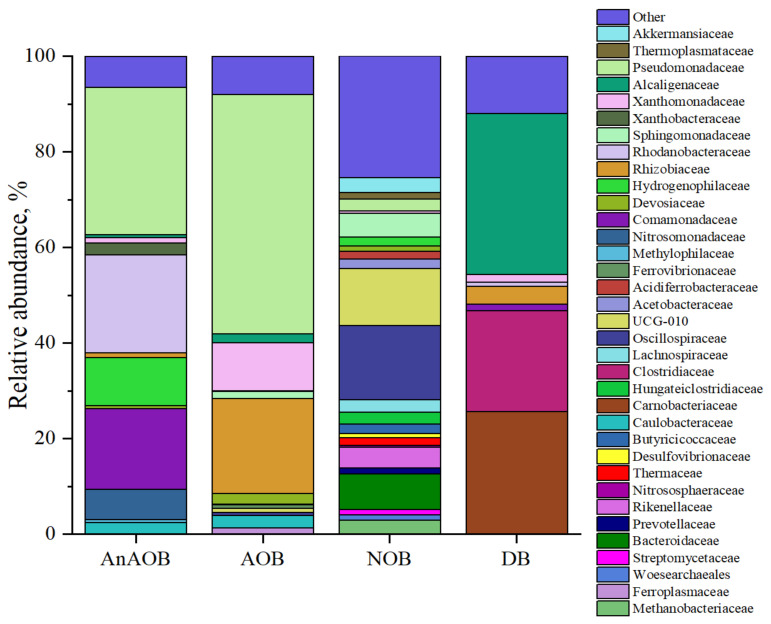
Microbial diversity of enrichment cultures in the batch experiment at the family level. AnAOB—anaerobic ammonium oxidizing (anammox) bacteria; AOB—aerobic ammonium oxidizing bacteria; NOB—nitrite oxidizing bacteria; DB—denitrifying bacteria.

**Figure 5 biology-12-00221-f005:**
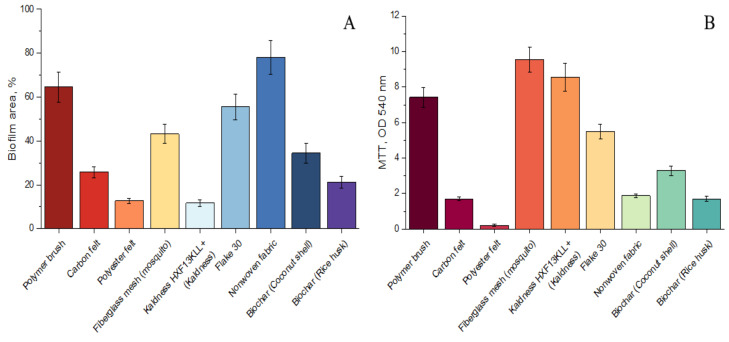
Area of carrier biofouling by microbial community, % according to confocal microscopy studies (**A**) and microbial respiration by MTT-assay (**B**), OD 540 nm.

**Figure 6 biology-12-00221-f006:**
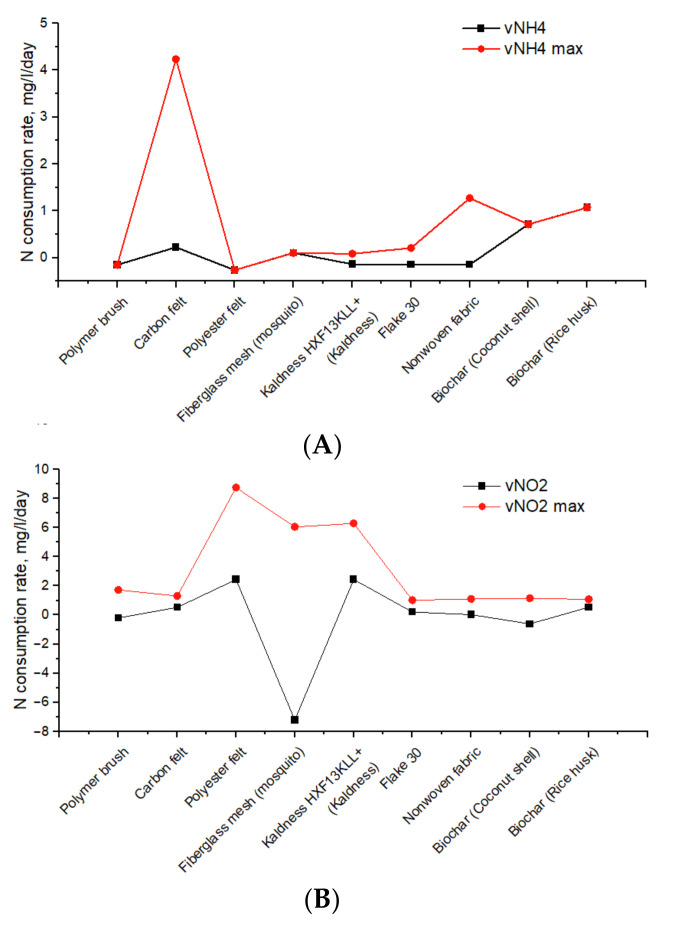
Ammonium (**A**) and nitrite (**B**) uptake rates in a batch experiment with the addition of various carriers.

**Figure 7 biology-12-00221-f007:**
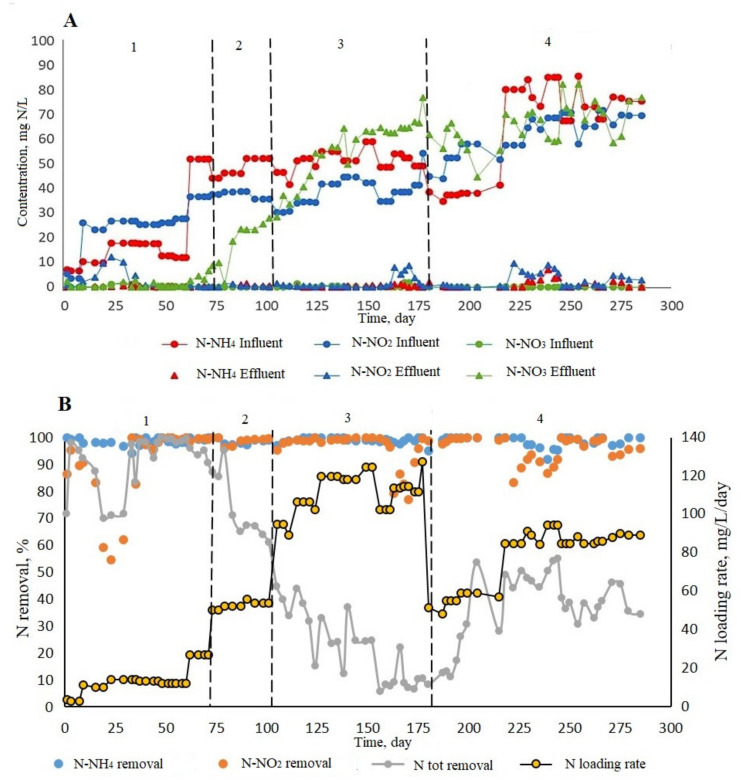
Time course of the N removal performance in the ABR. (**A**) Influent and effluent ammonium, nitrite, and nitrate nitrogen concentrations. (**B**) Ammonium, nitrate and total nitrogen (N tot) removal, and N loading rate.

**Figure 8 biology-12-00221-f008:**
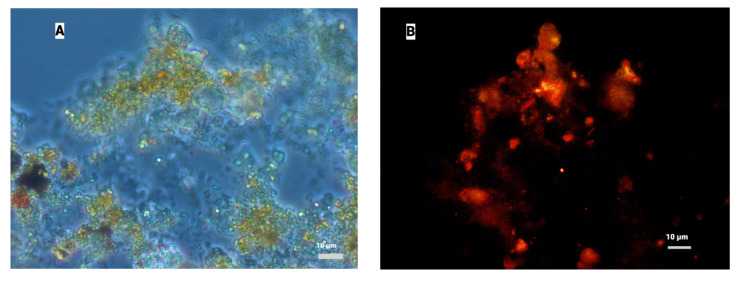
Hybridization of probes from phase 4 with Cy-3 labeled oligonucleotide probes, left—hybridization with probe, right—phase contrast: (**A**,**B**) amx368 (AnAOB-specific), (**C**,**D**)—Nsm156 (AOB-specific), (**E**,**F**)—Ntspa1151 (NOB-specific), (**G**) dynamics of the relative content of AnAOB, AOB, and NOB according to FISH probe data.

**Figure 9 biology-12-00221-f009:**
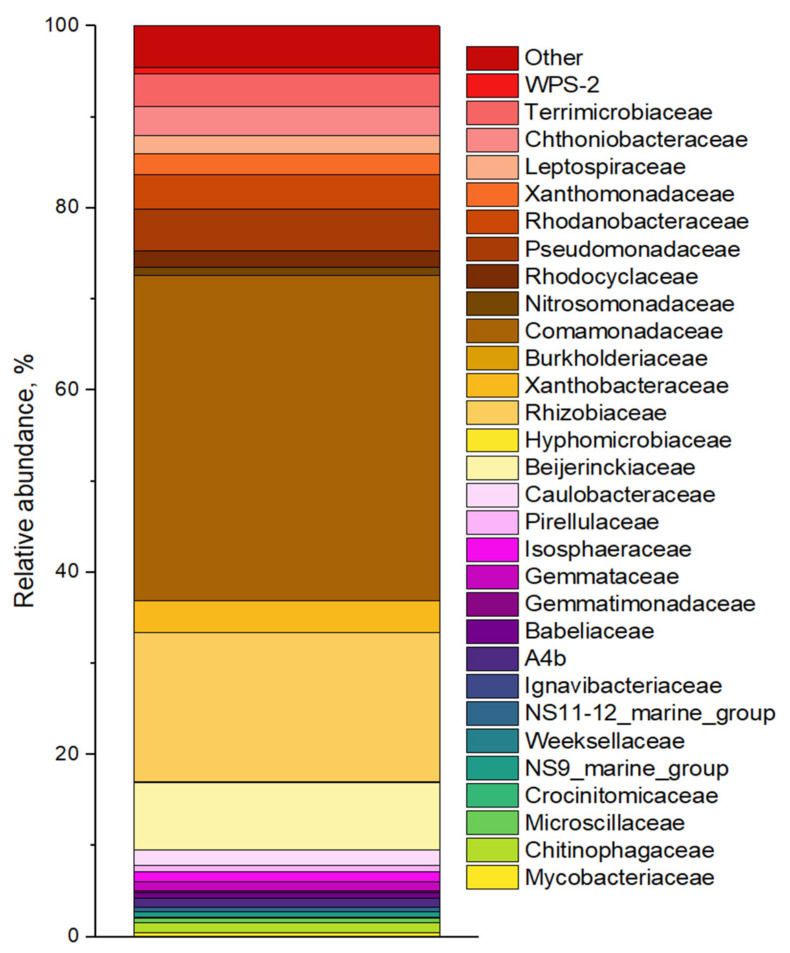
Stacked bar plot showing the average relative abundance of bacterial OTUs in ABR.

**Figure 10 biology-12-00221-f010:**
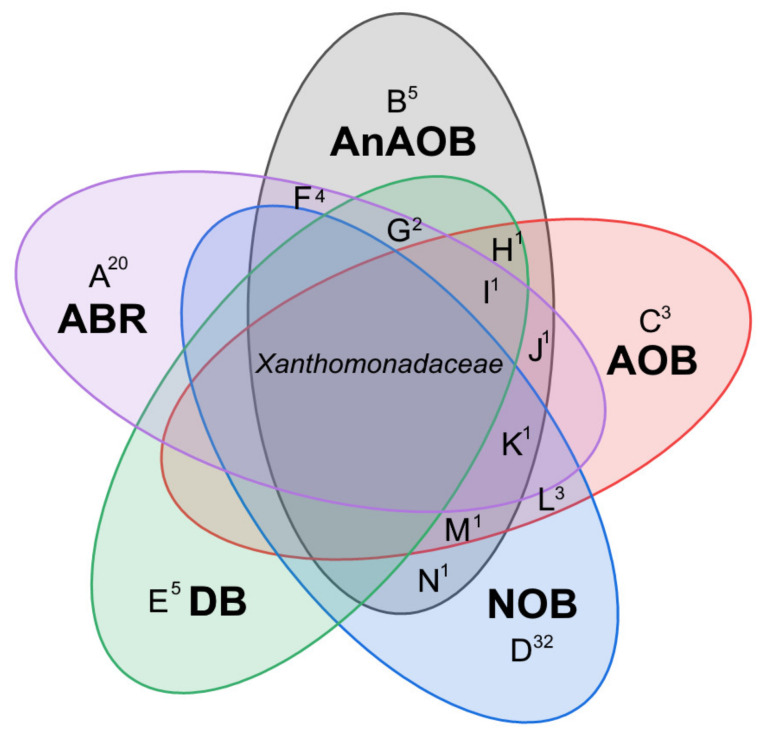
Distribution of bacterial taxa within enrichment cultures and the microbial community of ABR. Intersections are given in Appendix A, superscripts represent the number of OUT > 1% (A^20^-N^1^).

**Table 1 biology-12-00221-t001:** Chemical composition of the groundwater [33].

Parameter	Value
NH_4_, mg/L	58.4
NO_2_, mg/L	28.18
NO_3_, mg/L	7434
HCO_3_, mg/L	244
K, mg/L	447.6
Na, mg/L	879.7
Mg, mg/L	37.6
Ca, mg/L	1310
Cl, mg/L	1213
SO_4_, mg/L	1803
Fe_tot_, mg/L	2.1
Mn, mg/L	15.7
PO_4_, mg/L	1.1
U, µg/L	256
C_org_, mg/L	2.5
pH	7.1
Eh, mV	119
T, °C	7.5

**Table 2 biology-12-00221-t002:** Carriers used for immobilization of microbial community in batch experiments using anammox medium.

Carrier Material	BET Surface Area, m^2^/g
Polymer brush (Volgodonsk, Russia)	1.5
Carbon felt (ZLWMQMD 001 store, China)	0.4259
Polyester felt (JSC MONTEM, Russia)	0.0536
Fiberglass mesh (mosquito) (Phifer micro mesh, Tuscaloosa, AL, USA)	0.003526
HXF13KLL+ (Kaldness) (Hel-X, Marktrodach, Bavaria, Germany)	~0.003 (955 m^2^/m^3^)
Flake 30 (Hel-X, Marktrodach, Bavaria, Germany)	~0.01 (5000 m^2^/m^3^)
Nonwoven fabric (Geolia, Taiwan, China)	~300
Biochar (Coconut shell) (Noname, China)	772

**Table 3 biology-12-00221-t003:** ABR working conditions.

Phases (Days)	N-NH_4_ in the Influent, mg/L	N-NO_2_ in the Influent, mg/L	рН	Daily Influent Flow Rate, mL/Day
1 (1–80)	from 5 to 50	from 5 to 40	7.5–8	122.4
2 (80–105)	40–50	30–40		244.8
3 (105–180)	50–60	30–45		489.6
4 (180–208)	35–40	45–60		244.8
4 (208–285)	75–85	60–70		

**Table 4 biology-12-00221-t004:** Oligonucleotide probes used for FISH, and their specificity.

Probe	Sequence, 5′-3′	Target Group	Reference
Amx368	CCT TTC GGG CAT TGC GAA	all anammox bacteria	[54]
Nsm156	TAT TAG CAC ATC TTT CGA T	*Nitrosomonas* spp., *Nitrosococcus* spp.	[55]
Ntspa1151	TTC TCC TGG GCA GTC TCT CC	Sublineage II of the genus *Nitrospira* sp.	[56]
Nit3	CCT GTG CTC CAT GCT CCG (competitor: CCT GTG CTC CAG GCT CCG)	*Nitrobacter* spp.	[57]
Ntspa662	GGA ATT CCG CGC TCC TCT (competitor: GGA ATT CCG CTC TCC TCT)	*Nitrospira* spp.	[58]

## Data Availability

Data is contained within the article or Appendix A.

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
