# Peer review of "Characterization of Enrichment Cultures of Anammox, Nitrifying and Denitrifying Bacteria Obtained from a Cold, Heavily Nitrogen-Polluted Aquifer"

_biology, 2023, doi:10.3390/biology12020221_

Round 1
Reviewer 1 Report
In this manuscript, the authors set up enrichment cultures of anammox, denitrifying and nitrifying bacteria in order to get insight into nitrogen cycling processes in low temperature, heavily nitrogen contamined groundwater. They combined a broad spectrum of methods – enrichment cultures, fluorescence in situ hybridization and quantitative PCR – to characterize the microbial groups involved in the different nitrogen cycling processes. However, since they used standard enrichment approaches, their findings might not directly relate to what is happening under natural conditions. For example, denitrifier enrichments were set up with multiple different organic carbon sources, while the original aquifer system is characterized by low organic carbon content. This should be considered more strongly in the discussion and interpretation of the results. The manuscript contains valuable results about processes in enrichment cultures but the direct connection to the natural system should be treated with caution.
The relationship between the observed process rates and the abundance of the respective microorganisms is not fully convincing. For example, nitrogen removal and nitrate production activity in the ABR is mostly ascribed to anammox bacteria, even though there is little evidence for their presence in sufficient abundances. Did the authors attempt to make calculations if the observed activities could be explained by the number of anammox cells present, assuming published anammox rates per cell?
The introduction strongly focuses on anammox bacteria, while in the results section (and also in the title of the manuscript), all the different groups of nitrogen cycling microorganisms also receive considerable attention. Given the fact that the anammox enrichments in the end showed the lowest rate of success or produced controversial results, the authors might consider broadening the scope of the introduction to make it better match the content of the whole study.
Specific comments
l. 42-44: This statement is very general. Please be more specific about which organisms could be involved, based on the findings of this work.
l. 45: Was there evidence for aerobic ammonia oxidation?
l. 46: What is the exclusive role of Xanthomonadaceae?
l. 125-126: Are these the concentration values at the time point of sampling (since temperature is only given as a range over time)? Please specify.
l. 129: What was the oxygen concentration of the groundwater?
l. 146: Does this primer set target bacteria or both bacteria and archaea? Please explain.
l. 154: What were the oxygen settings for the enrichments of AOA/AOB and NOB?
l. 165-173: It appears that the anammox enrichments were briefly exposed to ambient air upon feeding. To what extent could potential presence of oxygen have affected the success of the enrichments?
l. 227: How were relative abundances determined, using DAPI as counterstain?
l. 249: which samples were used for DNA extractions, please specify.
l. 282-285: Is this a relationship which the authors would have expected? According to the stoichiometry of nitrification, should the consumption of ammonium not equal the rate of nitrite production? Please explain.
l. 322: Was the absence of the canonical nitrifiers also confirmed by FISH analysis?
l. 451-452: How do the authors explain the discrepancy between the FISH results and the qPCR results regarding the presence of anammox bacteria?
l. 474: What is the unit here? Number of cells per…?
l. 477-481: This information is not clear. Please rephrase.
l. 498: It is more likely that qPCR but not FISH targets dead anammox cells.
l. 625-629: This point is difficult to address since no data of the original groundwater community are given, and incubation conditions deviated substantially from in situ conditions, as artifical media were used.
Figure 2: For FISH pictures, it is common to also show results obtained with a nonsense probe or no probe at all to confirm the specificity of the probe signal. Are such results also available for this analysis?
Figure 6A: The colour code is not explained.
Author Response
Dear Reviewer, we thank you for the comprehensive and positive review of our manuscript. The comments were very constructive, and we tried to address all of the concerns. Below are the responses point by point.
In this manuscript, the authors set up enrichment cultures of anammox, denitrifying and nitrifying bacteria in order to get insight into nitrogen cycling processes in low temperature, heavily nitrogen contamined groundwater. They combined a broad spectrum of methods – enrichment cultures, fluorescence in situ hybridization and quantitative PCR – to characterize the microbial groups involved in the different nitrogen cycling processes.
However, since they used standard enrichment approaches, their findings might not directly relate to what is happening under natural conditions. For example, denitrifier enrichments were set up with multiple different organic carbon sources, while the original aquifer system is characterized by low organic carbon content. This should be considered more strongly in the discussion and interpretation of the results.
Response: We agree with the reviewer, that the composition of organotrophic denitrifying enrichment cultures, grown on acetate and glucose cannot be directly compared with the composition of the microbial community of groundwater. However, in view of the fact that we plan to conduct field experiments on the removal of nitrates in situ and the creation of anaerobiosis to start anaerobic processes by introducing soluble cheap simple electron donors (acetate, sugar), as was done earlier by us [10.3389/fmicb. 2018.01985] and other researchers [10.2166/aqua.2018.194; 10.1016/j.chemosphere.2018.04.133], the data obtained may be useful for a preliminary assessment of the most active organotrophic denitrifies. It should also be added that representatives of the Pseudomonadaceae, Rhodanobacteraceae, Xanthomonadaceae, and Alcaligenaceae families found in enrichment culture are standard representatives of the reducing branch of the nitrogen cycle in groundwater. In the groundwater sample, the microbial composition of which is given in our previous work [10.3390/biology11101421], bacteria of the Pseudomonadaceae family found in a denitrifying enrichment culture predominated.
These considerations have been added to the Discussion section.
The manuscript contains valuable results about processes in enrichment cultures but the direct connection to the natural system should be treated with caution.
Response: thanks, this valuable comment has been reflected in the Conclusion section
The relationship between the observed process rates and the abundance of the respective microorganisms is not fully convincing. For example, nitrogen removal and nitrate production activity in the ABR is mostly ascribed to anammox bacteria, even though there is little evidence for their presence in sufficient abundances. Did the authors attempt to make calculations if the observed activities could be explained by the number of anammox cells present, assuming published anammox rates per cell?
Response: this is a good question, thanks. We could not find specific N removal rates per anammox cell in the literature. However, some considerations can be made based on reported N removal rates per g of biomass-VSS:
- 08 g N/gV SS/d and 0.05 g N/g VSS/d for ammonium and nitrite, respectively [10.1016/j.jenvman.2008.03.003]
- 70 mg NTOTg VSS−1 d−1 [1016/j.heliyon.2021.e08445]
- 594-0.638 g N/g VSS·d [10.1051/e3sconf/20184400179]
- 598 ± 0.026 gN2-N gVSS⁻¹ d⁻¹ [10.1016/j.biortech.2017.02.117]
- 504 ± 0.127 gN (gVSS d)−1and 0.411 ± 0.037 gN (gVSS d)−1 [10.1007/s42452-019-0514-4]
- 4 ± 3.9 mg N g−1biofilm-VSS d−1 [10.1021/es803284y]
Thus, 0.002 to 0.64 g N/g VSS/day can be removed by the anammox process. Considering an average removal of 55 mg N/L/day in late phase 4 and an estimate of 0.1-0.3 g biomass-VSS/L in ABR, the calculated removal rate of 1.8-5.5 g N/g VSS/day seems too high for a purely anammox process. This means that not only anammox but also other microbial groups of the nitrogen cycle contributed greatly to nitrogen removal. The elucidation and in-depth study of these nitrogen-removing microbial groups should be one of the directions of future work.
These considerations were reflected in Abstract, Discussion and Conclusion sections.
The introduction strongly focuses on anammox bacteria, while in the results section (and also in the title of the manuscript), all the different groups of nitrogen cycling microorganisms also receive considerable attention. Given the fact that the anammox enrichments in the end showed the lowest rate of success or produced controversial results, the authors might consider broadening the scope of the introduction to make it better match the content of the whole study.
Response: The introduction has been modified in accordance with this valuable comment.
Specific comments
- 42-44: This statement is very general. Please be more specific about which organisms could be involved, based on the findings of this work.
Response: We agree with Reviewer, this statement was removed
- 45: Was there evidence for aerobic ammonia oxidation?
Response: Yes, in an appropriate (AOB) enrichment culture.
- 46: What is the exclusive role of Xanthomonadaceae?
Response: This has been more clearly explained in the Discussion section.
- 125-126: Are these the concentration values at the time point of sampling (since temperature is only given as a range over time)? Please specify.
Response: Yes, these are the concentrations at the time of sampling. Clarification has been added to the article
- 129: What was the oxygen concentration of the groundwater?
Response: We did not measure the concentration of dissolved oxygen in groundwater samples taken from this well. However, later (the next sampling in half a year) 0.6 mg O2/L was determined from the nearest well.
- 146: Does this primer set target bacteria or both bacteria and archaea? Please explain.
Response: This primer set is universal, targeting both bacteria and archaea [doi:10.1111/1758-2229.12684; doi:10 .1128/mBio.00824-17; doi:10.1007/s10482-013-9927-z]
- 154: What were the oxygen settings for the enrichments of AOA/AOB and NOB?
Response: AOB and NOB enrichment cultures were aerated at a very low rate (about 1 ml/min, through a medical needle connected to an aquarium pump).
This was added in the Section 2.3.
- 165-173: It appears that the anammox enrichments were briefly exposed to ambient air upon feeding. To what extent could potential presence of oxygen have affected the success of the enrichments?
Response: Yes, between feedings, the enrichment culture was exposed to air for a short time (10–20 sec), after which it was thoroughly purged with argon. However, we think that the low anammox activity was not due to exposure to air, but to the lack of continuous flow conditions and incorrect composition of the medium (lack of some important components).
- 227: How were relative abundances determined, using DAPI as counterstain?
Response: The relative abundances were determined by direct count of microbial cells, using DAPI as counterstain and phase contrast microscopy.
This was clarified in Section 2.7.
- 249: which samples were used for DNA extractions, please specify.
Response: The clarification was added
- 282-285: Is this a relationship which the authors would have expected? According to the stoichiometry of nitrification, should the consumption of ammonium not equal the rate of nitrite production? Please explain.
Response: Yes, you are right. To avoid confusion, we have recalculated in the text the rates of consumption and production of various forms of N from the ionic form to the nitrogen form, as was done in the previous Table 5 (now Figure 2).
- 322: Was the absence of the canonical nitrifiers also confirmed by FISH analysis?
Response: Unfortunately, FISH analysis was not applied for this enrichment.
- 451-452: How do the authors explain the discrepancy between the FISH results and the qPCR results regarding the presence of anammox bacteria?
Response: We believe that the number of anammox cells could be overestimated due to the specifics of the method (direct counting of hybridized cells under a microscope, which could lead to overestimation). Also, due to the fact that anammox cells appeared in clusters, their number could be overestimated: in such densely packed structures it is not always easy to distinguish one cell from another and estimate the exact number of hybridized cells.
This was explained in Discussion section: “Results of qPCR correlate with the results of FISH analysis. However, the amount of anammox bacteria estimated by FISH was one magnitude higher than that for qPCR. Presumably, anammox bacteria which naturally occur in densely packed clusters, could be overestimated during the direct count on the microscope when analyzing FISH probes.”
- 474: What is the unit here? Number of cells per…?
Response: Added (gene copies/mL)
- 477-481: This information is not clear. Please rephrase.
Response: The information was rephrased.
- 498: It is more likely that qPCR but not FISH targets dead anammox cells.
Response: We agree with this statement, the manuscript was modified.
- 625-629: This point is difficult to address since no data of the original groundwater community are given, and incubation conditions deviated substantially from in situ conditions, as artifical media were used.
Response: We agree with the reviewer and excluded this assumption from the Conclusion.
Figure 2: For FISH pictures, it is common to also show results obtained with a nonsense probe or no probe at all to confirm the specificity of the probe signal. Are such results also available for this analysis?
Response: These results are available and were not included in the manuscript.
Figure 6A: The colour code is not explained.
Response: Done.
Reviewer 2 Report
1. Abstract, L31, Candidatus Scalindua should be in Italics.
2. Table 1 can be made into two columns. No need to give separate column for units.
3. Section 2.2, can extend the method section about the R, version, package (metaseq, phyloseq, ggplot or so) information.
4. The medium used for NOB AOB DB, with Nitrates, nitrite, Glucose and acetate 1g/L
5. Table 2, just two columns should look better, eg: Carrier material (Polymer brush, Volgodonsk, Russia) and BET surface area (m2/g).
6. L230, standard scheme is not a correct term to use also the notation “Syntol”, Russia.
7. Please discuss the method for nitrogen degradation rate calculations or method for the better understanding to the readers.
8. Would recommend to change Table 5 into Figure for demonstrating the result.
9. Figure 6 quality can be improved.
10. Table 6 can be avoided for the qPCR.
11. Conclusion can be shorten.
Author Response
Dear reviewer, we thank you for your constructive comments on our manuscript, which helped to improve this work. We tried to address all of the concerns. Below are the responses point by point.
- Abstract, L31, Candidatus Scalindua should be in Italics.
Response: Thank you. Corrected in the text. In such cases, only Candidatus should be in Italics
- Table 1 can be made into two columns. No need to give separate column for units.
Response: Done
- Section 2.2, can extend the method section about the R, version, package (metaseq, phyloseq, ggplot or so) information.
Response: We have used standard formulations to describe the method as in previous works (e.g., https://doi.org/10.3390/biology11101421)
- The medium used for NOB AOB DB, with Nitrates, nitrite, Glucose and acetate 1g/L
Response: No, glucose and acetate were only used for DB enrichment as outlined in section 2.3.
- Table 2, just two columns should look better, eg: Carrier material (Polymer brush, Volgodonsk, Russia) and BET surface area (m2/g).
Response: Done
- L230, standard scheme is not a correct term to use also the notation “Syntol”, Russia.
Response: Corrected
- Please discuss the method for nitrogen degradation rate calculations or method for the better understanding to the readers.
Response: The equation was added in Section 2.3.
- Would recommend to change Table 5 into Figure for demonstrating the result.
Response: Done
- Figure 6 quality can be improved.
Response: Done
- Table 6 can be avoided for the qPCR.
Response: Done, it has been moved to Supplementary
- Conclusion can be shorten.
Response: The Conclusion section has been rewritten and somewhat condensed.
Reviewer 3 Report
This study tried to enrich psychrophilic and psychrotolerant microbial communities related with the nitrogen cycle. Authors used batch cultivation for successful AOB, NOB, and DN enrichment; and continuous flow cultivation with the addition of nonwoven fabric for AnAOB. While some phenomena appeared, for example, as the HRT became shorter, nitrate was increased in effluent which resulted the degradation of the TN removal efficiency in ABR system. qPCR found that the level of AOB was relatively high, as well as nirK, nirS denitrifiers compared with anammox. In addition, 16S rRNA discovered that no “classical” nitrifiers were found instead of Xanthomonadaceae family. The theory calculation proposed that there might be a new type of anammox hypothetical stoichiometry, in which anammox bacteria towards the production of more nitrate and less gaseous N2 at low temperatures. Besides, the NOB was outcompete anammox, which also could convert nitrite to nitrate. Based on these, it was really difficult to determine which caused the increase of nitrate.
Another question I want to discuss with the author was that weather we can find some evidence from the community succession from four stages of ABR. Why anammox was decreased in stage II and III, while recovered in stage 4. How does nitrifying bacteria, especially, NOB enrichment in ABR system. Did the dissolved oxygen came from the synthetic medium which was purged by argon and stored in an argon atmosphere as proposed.
Although complex phenomenon was appeared in this low temperature environment, these questions were discussed with positive speculation in this manuscript, therefore, this study still have its value for shedding light on the metabolism of anammox bacteria at low temperatures.
Some minor suggestions
1. Fig 6A, there was no legend caption.
2. Fig 8, if there have bacterial OTUs of other three stages, it would be more referential value.
Author Response
Dear Reviewer, we thank you for the comprehensive and positive review of our manuscript. The comments were constructive, and we tried to address all of the concerns. Below are the responses point by point.
This study tried to enrich psychrophilic and psychrotolerant microbial communities related with the nitrogen cycle. Authors used batch cultivation for successful AOB, NOB, and DN enrichment; and continuous flow cultivation with the addition of nonwoven fabric for AnAOB. While some phenomena appeared, for example, as the HRT became shorter, nitrate was increased in effluent which resulted the degradation of the TN removal efficiency in ABR system. qPCR found that the level of AOB was relatively high, as well as nirK, nirS denitrifiers compared with anammox. In addition, 16S rRNA discovered that no “classical” nitrifiers were found instead of Xanthomonadaceae family. The theory calculation proposed that there might be a new type of anammox hypothetical stoichiometry, in which anammox bacteria towards the production of more nitrate and less gaseous N2 at low temperatures. Besides, the NOB was outcompete anammox, which also could convert nitrite to nitrate. Based on these, it was really difficult to determine which caused the increase of nitrate.
Another question I want to discuss with the author was that weather we can find some evidence from the community succession from four stages of ABR. Why anammox was decreased in stage II and III, while recovered in stage 4. How does nitrifying bacteria, especially, NOB enrichment in ABR system. Did the dissolved oxygen came from the synthetic medium which was purged by argon and stored in an argon atmosphere as proposed.
Response: Thank you for these good questions. A possible explanation for this observation is given in the Discussion section. In particular, we believe that the abundance of anammox bacteria could have decreased due to suboptimal cultivation conditions, for example, the composition of the synthetic medium differed from the composition of groundwater and some necessary elements could be missing. However, as they were cultivated, they were able to adapt to new conditions. As for nitrifying bacteria, they could thrive in the system with using the remaining DO in the medium after purging with argon. At least similar results on the high build-up of nitrates in the anaerobic system of the anammox reactor have been reported by Li et al [10.1016/j.watres.2019.115279].
Although complex phenomenon was appeared in this low temperature environment, these questions were discussed with positive speculation in this manuscript, therefore, this study still have its value for shedding light on the metabolism of anammox bacteria at low temperatures.
Some minor suggestions
- Fig 6A, there was no legend caption.
Response: Corrected
- Fig 8, if there have bacterial OTUs of other three stages, it would be more referential value.
Response: Unfortunately, only the last phase was sampled for 16S genomic analysis.
Round 2
Reviewer 1 Report
In the revised version of this manuscript, the authors addressed all my previous comments, and the manuscript has been improved substantially.
I have only a few comments specifically to the discussion and conclusions:
l. 647-650: This is highly speculative. First, the assimilatory nitrate reduction described here might not be relevant for the N transformations observed in the ABR. Second, the current state of knowledge is that nitrate reduction to ammonium, either assimilatory or dissimilatory, requires a couple of enzymatic steps via the intermediate nitrite.
l. 659-661: But would those taxa not be limited by oxygen availability either? It was stated earlier in this manuscript that nitrate formation in the ABR could not fully be explained by activity of AOB and NOB because their oxygen demands would be too high. Would the same limitation not apply to other taxa harbouring NXR?
l. 685-687: I would suggest being more cautious with this statement. The absence of canonical nitrifiers only refers to the AOB and NOB enrichments. AOB and NOB were indeed enriched in the ABR system, and no data about the original groundwater community are provided to demonstrate their absence under in situ conditions.
Specific comments:
l. 45: replace “microbial communities” by “enrichment cultures”
l. 67: delete “is”
Author Response
In the revised version of this manuscript, the authors addressed all my previous comments, and the manuscript has been improved substantially.
Response: Once again, we thank reviewer for his genuine interest in our work and very comprehensive comments. Below are the responses point by point.
I have only a few comments specifically to the discussion and conclusions:
l. 647-650: This is highly speculative. First, the assimilatory nitrate reduction described here might not be relevant for the N transformations observed in the ABR. Second, the current state of knowledge is that nitrate reduction to ammonium, either assimilatory or dissimilatory, requires a couple of enzymatic steps via the intermediate nitrite.
Response: The reviewer is right; in this case we have no direct evidence of assimilatory nitrate reduction by qPCR assays or by biochemical evaluation of enzymatic activity. On the other hand, the assimilatory nitrate reduction can proceed more slowly than the dissimilatory nitrate reduction, and the accumulation of nitrites will not be so significant. In addition, there are works reporting that high concentrations of ammonium can reduce the intensity of dissimilatory nitrate reduction, while assimilatory route can proceed [https://doi.org/10.1016/j.biortech.2020.123597].
We have changed the text “In the Xanthomonadaceae family, according to the analysis of 12 known strains in the KEGG database, an assimilation pathway for the reduction of nitrate to ammonium was found. Perhaps they reduce nitrate without the stage of nitrite formation. In the case of high nitrate concentrations, this feature alleviates toxic stress of nitrite, in contrast to classical denitrifiers.”
to the following:
“In the Xanthomonadaceae family, according to the analysis of 12 known strains in the KEGG database, an assimilation pathway for the reduction of nitrate to ammonium was found. In this case, the accumulation of nitrite may not be as significant due to the lower rate of the process. In the presence of high concentrations of nitrate, this feature alleviates toxic stress of nitrite, in contrast to classical denitrifiers.”
l. 659-661: But would those taxa not be limited by oxygen availability either? It was stated earlier in this manuscript that nitrate formation in the ABR could not fully be explained by activity of AOB and NOB because their oxygen demands would be too high. Would the same limitation not apply to other taxa harbouring NXR?
Response: In a work of Li et al. (2020), the detected average dissolved oxygen concentration in anaerobic reactors were less than 0.1 mg/L, while (1) metagenomic analysis showed that relatively higher NXR levels correlate both with higher-than-expected levels of nitrate production, and with a decrease in the number of anammox bacteria. Also, in situ assays suggested that the higher nitrate production was not solely due to obligate NOB, but other nxr-containing bacteria are important contributors as well [doi: 10.1016/j.watres.2019.115279]. In addition, members of both of these genera (Thermomonas and Stenotrophomonas) are known as facultative or strictly aerobic bacteria. However, they can anaerobically assimilate substrates in the absence of in situ electron acceptor addition, indicating unknown anaerobic activity [doi: 10.1111/1462-2920.12614], or they are isolated from a laboratory anaerobic upflow sludge blanket reactor [doi: 10.1099/00207713-52- 2-559]. So, at the moment, we cannot fully understand the role of the main participants of enrichment cultures in the N cycle, further study is needed. Perhaps there is a more complex N cycle than we realize, or some hitherto undescribed processes. We can only say that based on the high representation of these groups, it is likely that they play some key roles in N transformation processes.
l. 685-687: I would suggest being more cautious with this statement. The absence of canonical nitrifiers only refers to the AOB and NOB enrichments. AOB and NOB were indeed enriched in the ABR system, and no data about the original groundwater community are provided to demonstrate their absence under in situ conditions.
Response: The composition of the microbial community of groundwater is given in our previously published paper [doi: 10.3390/biology11101421]. In particular, we found members of the Devosiaceae family capable of nitrification and the subsequent reduction of nitrite to nitric oxide during denitrification and nitrite reduction to ammonium and members of the genus Aeromonas (family Aeromonadaceae) performing heterotrophic nitrification and aerobic denitrification. The presence of nitrifiers was not detected.
We have made changes in the text. Instead of
l. 337-338: “Surprisingly, “canonical” nitrifiers were not found in the microbial communities.”
Now we write
l. 333-335: “Canonical” nitrifiers were not found in the microbial communities, as well as in the original groundwater [33].
Specific comments:
- 45: replace “microbial communities” by “enrichment cultures”
Response: Done
- 67: delete “is”
Response: Done